# Dynamic Relocation in Ridesharing via Fixpoint Construction

**Ian A. Kash**[1]        **Zhongkai Wen**[1]        **Lenore D. Zuck**[1]

[1]University of Illinois Chicago, Chicago, IL, USA

## Abstract

To address spatial imbalances in the supply and demand of drivers, ridesharing platforms can make use of policies to direct driver relocation. We study a simple model of this problem, which allows us to give a constructive characterization of the unique fixpoint of system dynamics. Using this construction, we design a dynamic policy that provides stronger, than previous work, guarantees about its rate of convergence to the fixpoint. Simulations demonstrate the benefits of our approach.

## 1 INTRODUCTION

Ridesharing platforms such as Didi, Lyft, and Uber match passengers in need of transportation with drivers who can provide it. As drivers provide service to passengers, they themselves travel and so may end up in a new region when seeking to next provide service. Unbalanced demand to and from a region leads to an excess or shortage of drivers in that region and calls for a policy for *relocation* of drivers. Designing good relocation policies that direct drivers from regions where there is an excess to regions where there is a shortage is therefore a key challenge both for providing efficient service [Afeche et al., 2018] and minimizing environmental costs [Ward et al., 2021].

Prior work has developed a number of models of the problem of designing relocation policies, including in the context of other decisions such as pricing. Many of these models are, however, quite complex and the results are often focused on optimizing a particular objective such as revenue or availability. This leads to several related challenges. In some analyses it is simply assumed that the system has achieved some notion of steady state or equilibrium without justification of how the system dynamics lead there [Bimpikis et al., 2019, Besbes et al., 2021]. Furthermore, this steady-state behavior may only be examined for an optimal policy for a particular objective, leaving the behavior of the system if a policy is chosen to address another objective [Bimpikis et al., 2019, Hosseini et al., 2021, Iglesias et al., 2019, Zhang and Pavone, 2016]. Even for those few analyses that have addressed the convergence behavior of arbitrary policies, the complexity of the model has typically lead to a non-constructive analysis [Braverman et al., 2019].

In this paper we address these problems by adopting a simple stylized model that can be thought of as a special case or limiting behavior of a number of models in the literature (see Section 1.1). In it, the area where the ridesharing platform operates is divided into $r$ regions and there is a fixed total mass of drivers available to serve passengers. Time is discrete and at each step a fixed mass of passengers seek service in each region. The drivers currently in that region carry passengers to their destination. If the mass of drivers exceeds the mass of passengers in a region, the remaining drivers relocate according to a fixed policy (which may include staying where they are). Both carrying passengers and relocating take a single time unit. If the mass of passengers exceeds the mass of drivers in a region, the excess passengers are simply not served.

The simplicity of our model allows us to provide concise arguments that establish a number of key properties.

- Each combination of policy and total mass of drivers has a unique fixpoint of the system dynamics.

- Starting from any initial conditions, the system dynamics converge to this unique fixpoint.

- The fixpoint is continuous and monotone in the mass of drivers and can be constructed via the stationary distributions of a linear number of Markov chains.

Our arguments are based on the analysis of a piecewise linear generalization of Markov chains which may be of independent interest.

As an example of the benefits of such a rigorous understanding of the behavior of the system, we analyze the problem

*Accepted for the 38th Conference on Uncertainty in Artificial Intelligence* (UAI 2022).

of dynamically adjusting the policy to converge to the fix-point as rapidly as possible from arbitrary initial conditions. We introduce a dynamic policy that makes use of our fix-point construction and is the first to have guarantees about its rate of convergence and the welfare loss while converging relative to welfare in the chosen fixpoint. Simulations based on data from Didi show superior performance to prior approaches for handling stochastic demands. Additional simulations on synthetic data show that it converges substantially faster than previous heuristic approaches. In our simple model these previous approaches target maximizing efficiency and our approach matches their performance while being more flexible in its ability to target other metrics such as availability.

## 1.1 RELATED WORK

Our model is part of a growing literature that analyzes the stationary behavior of ridesharing systems either in a direct formulation as optimization problem [Bimpikis et al., 2019, Pavone et al., 2012] or as the fluid limit of a queueing model [Braverman et al., 2019, Hosseini et al., 2021, Iglesias et al., 2019, Banerjee et al., 2017, Zhang and Pavone, 2016, Waserhole and Jost, 2012]. Indeed, our model is a special case of a number of these. There are also models of service networks which lack the crucial aspect of ridesharing that providing service changes which customers can be served in the future [Caldentey et al., 2009, Adan and Weiss, 2012, Gurvich and Ward, 2014].

Closest to our theoretical results, Braverman et al. [2019] analyze the dynamics of a queueing system with a fluid limit, which generalizes our model. As we do, they show convergence to a unique fixpoint. Their analysis is non-constructive and substantially more complex. They examine dynamic relocation but provide only simple heuristics. Our analysis is constructive and simpler, and we provide theoretical guarantees for dynamic relocation.

Hosseini et al. [2021] recently examined the use of dynamic policies to address stochastic deviations from the fixpoint of system dynamics due to a finite population of drivers and passengers. While their analysis is quite different from ours, they exploit some of the same underlying properties of the dynamics and we use an adaptation of their heuristic as a comparison in our experiments. As we demonstrate, in our experiments we obtain faster convergence and can target a wider range of objectives.

While our focus is on the design of relocation policies, there is also a literature on the implications of implementing them with self-interested drivers through pricing or other mechanisms. Closest to our work are those on spatial pricing [Afeche et al., 2018, Besbes et al., 2021, Lu et al., 2018, Bimpikis et al., 2019], but other work examines temporal policies such as charging "surge" prices at times of peak demand [Hall et al., 2015, Banerjee et al., 2015, Cachon et al., 2017, Chen and Sheldon, 2015, Garg and Nazerzadeh, 2021, Hall et al., 2017] and combining both spatial and temporal pricing [Buchholz, 2015, Guda and Subramanian, 2017, Ma et al., 2019].

While we treat all drivers and passengers in a region as interchangeable, there are also finer-grained models which focus on decisions about which specific driver to match to each passenger [Hu and Zhou, 2022, Castillo et al., 2017, Özkan and Ward, 2020, Biswas et al., 2017].

## 2 PRELIMINARIES

### 2.1 NOTATION

Let $r$ denote the number of regions. In the sequel, all the vectors are $r$-dimensional and all the matrices are $r \times r$ dimensional. We sometimes abuse notation and denote by $A[i, j]$ the term $(A[i])[j]$.

For two vectors $A$ and $B$, we say that $A \geq B$ if $\max(A, B) = A$, where the $\max$ (here and elsewhere in this document) is taken point-wise. Thus, for every $i$, $A[i] \geq B[i]$. Similarly, for vectors $A$ and $B$, $A \leq B$ if $\min(A, B) = A$. Similarly, if $A$ and $B$ are matrices, we say that $A \geq B$ (resp. $A \leq B$) if for every $i$ and $j$, $A[i, j] \geq B[i, j]$ (resp. $A[i, j] \leq B[i, j]$).

We denote by $\mathbb{c}$ the vector whose entries are all $c$. In particular we use $\mathbb{0}$—the all 0 vector, and $\mathbb{1}$—the all 1 vector.

We use boldface to denote the sum of a vector or a matrix. Thus, $\mathbf{A}$ is the sum of $A$'s entries.

For a matrix $A$, $\mathbb{1}A$ is the vector whose $i^{th}$ entry is the sum of the $i^{th}$ column of $A$. Similarly, $A\mathbb{1}$, is the vector whose $i^{th}$ entry is the sum of the $i^{th}$ row of $A$.

For a positive real $x$, let a *vector induced by $x$* be any ($r$-dimensional) vector of non-negative reals whose sum is $x$. We call a vector $A$ induced by 1 a *probability vector*. Every vector induced by a positive $x$ uniquely defines a probability vector $A$ by normalization. Conversely, every probability vector $A$ defines a vector induced by $x$, namely $xA$, which we call the *the vector induced by $x$ given $A$*.

A *right stochastic matrix*, or just a *stochastic matrix* is a matrix all of whose rows are probability vectors. Such a matrix corresponds to a Markov chain and is *ergodic* if for some power $k$, $A^k$ is positive. Ergodic matrices have unique *stationary distribution*. We use the predicate $Erg(A)$ to denote that a stochastic matrix $A$ is ergodic. If $Erg(A)$, we denote by $\sigma(A)$ its unique stationary distribution, thus $\sigma(A) = \sigma(A)A$.

## 2.2 MODEL

Our model of a ridesharing system has four key parts:

1. A demand matrix $W$ such that $W[i,j]$ denotes the mass of passengers wishing to travel from $i$ to $j$. For technical convenience we assume that for every $i$ and $j$, $W[i,j] > 0$. $W$ induces a right stochastic matrix which we denote $V$ where $V[i,j]$ is the fraction of passengers in region $i$ who wish to travel to $j$;

2. A policy $\pi$—a probability vector where $\pi[j]$ denotes the fraction of drivers who have no passengers that relocate to region $j$. We denote by $Z_\pi$ the set of regions $i$ for which $\pi[i] = 0$. For notational convenience we sometimes refer to a stochastic matrix $\Pi$ where each row is $\pi$.

3. A total mass of drivers in the system $q \geq 0$.

4. At each timestep the current state of the system is represented by a mass distribution $M$ where $M[i]$ denotes the mass of drivers at region $i$. Thus we require $M \geq \mathbb{0}$ and $\mathbf{M} = q$.

From a region $i$, the first up to $W\mathbb{1}[i]$ drivers are distributed *full*, that is with passengers, to every region $j$ proportional to $V[i,j]$. The other drivers, if any, drive *empty*, that is passenger-less, to a destination according to $\pi$. A region $i$ for which $M[i] \geq W\mathbb{1}[i]$ is called *saturated*; otherwise it is called *unsaturated*. Let $U_M$ be the set of unsaturated regions given $M$. Let $out(M)[i]$ be the vector describing the outflow from region $i$ given $M$. For every region $i$,

$$out_{W,\pi}(M)[i] = \begin{cases} M[i]V[i] & i \in U_M \\ W\mathbb{1}[i]V[i] + (M[i] - W\mathbb{1}[i])\pi & \text{o.w.} \end{cases}$$

The first line, as well as the first term in the second line, refer to full rides from $i$, while the second term in the second line refers to empty rides from $i$.

We assume that time is discrete and all rides and relocations take unit time. Denote by $next(M)$ the vector of the drivers available in each location in the next step. Then for every region $i$,

$$next(M)_{W,\pi}[i] = \sum_j out_{W,\pi}(M)[j,i]$$

For brevity, when it does not create ambiguity we may omit the $W$ or $\pi$ from the subscript. A fact that will be used repeatedly is that $next(M)$ (and its component parts) are monotone and continuous in $M$:

**Lemma 1** *The functions $out(M)$ and $next(M)$ are continuous and increasing in $M$.*

**Proof:** Consider a region $i$ and the function $out(M)[i]$ which is either a product of $M[i]$— a scalar which is continuous in $M$—and a constant vector, or a sum of two such

vectors. Hence $out(M)[i]$ is continuous. In either case, it is also increasing in $M$. Since $next(M)$ is a summation over $out(M)$'s columns, the claim follows. $\bowtie$

The stylized model we have described is a special case of a number of models in the literature (e.g. [Bimpikis et al., 2019, Braverman et al., 2019, Hosseini et al., 2021]). We have made it as simple as possible to enable a clear, concise analysis. We discuss relaxing assumptions in Sec. 8.

## 3 CONSTRUCTING A FIXPOINT

Fix a demand matrix $W$ and a policy $\pi$. We present an algorithm (Alg. 1) that, given a quantity $q \geq 0$, constructs a mass distribution $M$ such that $\mathbf{M} = q$ and $M = next(M)$. Here, $next(M)$ is analogous to the action of a Markov chain and such a fixed point would correspond to a stationary distribution. However, in general the action of $next(M)$ is not a Markov chain because $out_{W,\pi}$ is piecewise linear rather than linear. We first introduce some notation and claims that are used in the algorithm and its proof of correctness.

### 3.1 BUILDING BLOCKS

A mass distribution $M$ defines the *marginal transition matrix* $T_M$ where for every region $i$,

$$T_M[i] = \begin{cases} V[i] & i \in U_M \\ \pi & \text{otherwise} \end{cases}$$

Because $W[i,j] > 0$ for all $i$ and $j$, $T_M$ is ergodic for every $M$ for which $U_M$ includes some non-$Z_\pi$ regions. Given $M$ such that $Erg(T_M)$ and $\sigma = \sigma(T_M)$, define

$$q(M) = \min_{i \in U_M} ((W\mathbb{1}[i] - M[i])/\sigma[i])$$

That is, $q(M)$ is the maximal mass that can be added, according to $\sigma$, to $M$ as not to cause any region that was unsaturated to be oversaturated. The following lemma establishes that if $M = next(M)$ and there are some non-$Z_\pi$ unsaturated regions, then adding any quantity $q \leq q(M)$ to $M$, proportionally to $\sigma(T_M)$, will maintain the fixpoint.

**Lemma 2** *Assume $M = next(M)$ and $U_M \cap \overline{Z_\pi} \neq \emptyset$, and let $\sigma = \sigma(T_M)$. Then for every $q'$, $0 \leq q' \leq q(M)$, $M + q'\sigma$ is a fixpoint mass distribution of $next()$.*

**Proof:** Consider some $q'$, $0 \leq q' \leq q(M)$ and $i \in U_M \cap \overline{Z_\pi}$. The additional outflow from region $i$ in $M' = M + q'\sigma$ relative to $M$ is $q'\sigma[i]$. Every region $j$ contributes $q'\sigma[j]T_M[j,i]$ into $i$. Since $\sigma = \sigma(T_M)$, $\sum_j q'\sigma[j]T_M[j,i] = q'\sigma[i]$. $\bowtie$

The following lemma establishes that if $M = next(M)$ with all unsaturated regions in $Z_\pi$, then adding any quantity $q \geq 0$ to $M$, according to $\pi$, will maintain the fixpoint.

**Lemma 3** *Assume $M = next(M)$ and $U_M \subseteq Z_\pi$. Then for every $q \geq 0$, $M + q\pi$ is a fixpoint mass distribution.*

The proof, along with subsequent omitted proofs, can be found in App. C

### 3.2 THE CONSTRUCTION

Let $q \geq 0$. We now describe a construction that allocates $q$ drivers into a mass distribution $M$ such that $M$ is a fixpoint. The function mapping $q$ into the fixpoint is piecewise linear and is accomplished in at most $r + 1$ phases. In all but possibly the last phase, a portion of the remaining $q$-allocation is distributed among the regions as to satisfy the fixpoint yet so that no unsaturated region becomes oversaturated, until either $q$ is exhausted or all non-$Z_\pi$ regions are saturated. If all non-$Z_\pi$ regions are saturated and $q$ is not exhausted, the remaining mass is distributed according to $\pi$. The construction algorithm is in Alg. 1.

---
**Algorithm 1:** Construction of Fix Point
---
**Input:** A willing demand $W$, a policy $\pi$, and a total mass of drivers, and $q$
**Output:** A mass distribution $M$ such that $\mathbf{M} = q$ and $next(M) = M$
1  $M$ : a vector, **init** $0$ /* current mass distribution                                      */
2  $m$ : **init** $q$ /* current mass                     */
3  $i$ : **init** $r$ /* ghost variable for correctness proof                        */
4  **while** $(m > 0)$ **do**
5      **if** $U_M \subseteq Z_\pi$ **then**
6          $\sigma \leftarrow \pi$
7          $\Delta q \leftarrow m$
8      **else**
9          $\sigma \leftarrow \sigma(T_M)$
10         $\Delta q \leftarrow \min(q(M), m)$
11     **end**
12     $M \leftarrow M + \Delta q \cdot \sigma$
13     $m \leftarrow m - \Delta q$
14     $i \leftarrow i - 1$
15 **end**
16 **return** $M$

---

We now show that:

**Theorem 1** *The Algorithm in Alg. 1 terminates, and upon termination $M = next(M)$ and $\mathbf{M} = q$.*

## 4 UNIQUENESS OF FIXPOINT FOR $next$

Fix a demand matrix $W$ and a policy $\pi$. For every $q \in \mathbb{R}^+$ we have shown how to construct a fixpoint of $next$ with total mass $q$. We now show this fixpoint is unique. Let

$$S_q = \{M \in (\mathbb{R}_{\geq 0})^r : \mathbf{M} = q\}$$

denote the set of mass distributions with total mass $q$. In the proof, rather than confining our analysis to $S_q$ we extend it to a complete lattice $\mathcal{L}$ on which one can, with the aid of an auxiliary monotonically increasing function ($aux$) that has the same fixpoints as $next$, apply the Knaster-Tarski theorem to show that the fixpoint of $next$ in $S_q$ is unique.

**Theorem 2** *Let $W$ and $\pi$ be given. For all $q \geq 0$, the function $next$ has a unique fixpoint in $S_q$.*

The proof of the theorem relies on the following technical lemma. Consider two mass distributions $M_0$ and $M_1$ such that $M_0 \lesssim M_1$ and let $i$ and $j$ be regions such that $next(M_0)[i] = next(M_1)[i]$ and $M_0[j] < M_1[j]$. That is, the increase in mass from $M_0$ to $M_1$ adds drivers to $j$ but does not result in additional inflow to $i$. The following lemma show that this is equivalent to having $i$ assigned no rides by $\pi$ and $j$ being saturated in $M_0$.

**Lemma 4** *Let $M_0$ and $M_1$ be mass distribution vectors such that $M_0 \lesssim M_1$. Let $J \subseteq U_{M_0}$ be the set of unsaturated regions $j$ for which $M_0[j] < M_1[j]$. Then for all regions $i$,*

$$next(M_0)[i] = next(M_1)[i] \quad \textit{iff} \quad i \in Z_\pi \textit{ and } J = \emptyset$$

## 5 OPTIMAL MASS ALLOCATION

So far, we have shown how given a fixed willingness $W$ (and thus also $V$), policy $\pi$, and mass $q$ we can compute the unique fixpoint. However, a ridesharing platform will typically have at least some control over $\pi$. A natural question is how $\pi$ should be chosen. We examine how this can be done to maximize the number of full rides. We give a linear programming approach to calculating such an optimal $\pi$. This type of approach has been used in a number of similar models [Braverman et al., 2019, Hosseini et al., 2021, Bimpikis et al., 2019], but we provide a complete treatment as we use it in our experiments

Recall the definition of *out* from Sec. 2. There we split the computation of number of rides outgoing from a region according to whether the region is saturated. If the region is unsaturated, then obviously all outgoing rides are full. Else, some outgoing rides are empty, and distributed according to $\pi$. Let $F[i, j]$ denote the full outgoing rides from $i$ to $j$ and $E[i, j]$ the empty outgoing rides. That is, for a mass distribution $M$,

$$F(M)[i] = \begin{cases} M[i]V[i] & M[i] < W\mathbb{1}[i] \\ W[i] & \text{otherwise} \end{cases}$$

and

$$E(M)[i] = \begin{cases} 0 & M[i] < W\mathbb{1}[i] \\ (M[i] - W\mathbb{1}[i])\pi & \text{otherwise} \end{cases}$$

Rather than directly calculating a policy $\pi$ that at the fixpoint maximizes $\boldsymbol{F}$, we instead provide a linear program in Fig. 1 that defines when a solution $\langle F, E \rangle$ is feasible in that it results from the fixed point of some policy $\pi$ for supply $q$. The first constraint restricts to non-negative values. The second, **S**upply-constraint, requires that the total traffic is $q$. The third, **F**low-constraint, requires that at each region the in- and out- flows are equal (thus the solution is a fixpoint). The fourth, **D**emand-constraint, requires that the full outgoing traffic at each region is no more than the willing demand at the region. Finally, the fifth, **P**roportion-constraint, requires that full outgoing traffic at each region is allocated according to the willingness entry for that region.

| | |
|---|---|
| $F[i,j], E[i,j] \geq 0 \ \forall i, j$ | |
| $(\boldsymbol{F} + \boldsymbol{E}) = q$ | S-constraint |
| $(F + E)\mathbb{1}[i] = \mathbb{1}(F + E)[i] \ \forall i$ | F-constraint |
| $F\mathbb{1}[i] \leq W\mathbb{1}[i] \ \forall i$ | D-constraint |
| $F[i,j] = F[i,k] \cdot V_W[i,j]/V_W[i,k] \ \forall i,j,k$ | P-constraint |

Figure 1: Definition of feasibility of $\langle F, E \rangle$ for supply $q$

Below are some properties of feasibility as defined here.

**Observation 1** *Let $W$ and $\pi$ be given. Given any $q \geq 0$, there is a feasible solution for $q$.*

**Observation 2** *Let $W$ and $\pi$ be given and let $\langle F, E \rangle$ be feasible for $q$. Then the following all hold:*

1. *For every $c$, $0 \leq c \leq 1$, $\langle cF, cE \rangle$ is feasible for $cq$;*

2. *If $\langle F', E' \rangle$ is feasible for $q'$ and $(F + F') \leq W$, then $\langle F + F', E + E' \rangle$ is feasible for $q + q'$;*

3. *(Every non-trivial $x > 0$ carrying empty cycle can be removed:)*

    (a) *If for some region $i$, $E[i,i] > x > 0$, then if $E'$ is just like $E$ only that $E[i,i] = E[i,i] - x$, then $\langle F, E' \rangle$ is feasible for $q - x$;*

    (b) *If for some regions $i$ and $j$, $i \neq j$, and some $x > 0$, $E[i,j], E[j,i] \geq x$, then if $E'$ is just like $E$ only that $E'[i,j] = E[i,j] - x$ and $E'[j,i] = E[j,i] - x$, then $\langle F, E' \rangle$ is feasible for $q - 2x$.*

The requirements of feasibility are linear, thus the problem OA, in Fig. 2, is a linear programming problem.

| |
|---|
| Maximize $\boldsymbol{F}$ such that $\langle F, E \rangle$ is a feasible solution for supply $q$ |

Figure 2: The Optimal Allocation (OA) problem

The only requirement of a fixpoint not directly enforced by feasibility is that drivers do not leave empty if there are

passengers waiting. The following lemma shows that, at the optimal solution, every region has a greedy strategy: it only sends empty cars after all demand is satisfied.

**Lemma 5** *Let $q \geq 0$, and assume $\langle F_0, E_0 \rangle$ is feasible for $q$. If there exists some $i_0$ and $j_0$ such that $E_0[i_0, j_0] > 0$ and $F_0[i_0, j_0] < W[i_0, j_0]$, then $\langle F_0, E_0 \rangle$ is not an optimal solution of OA.*

# 6 DYNAMIC RELOCATION VIA FIXED POINT CONSTRUCTION

In this section we introduce our formulation of the problem of computing a dynamic relocation policy, introduce a particular policy based on our fixed point construction in Alg. 1, and show it has attractive theoretical properties in terms of convergence rate and welfare loss while converging.

## 6.1 THE DYNAMIC RELOCATION PROBLEM

Given $W$ and $\pi$ and some $q > 0$, let $M^*$ such that $\mathbf{M}^* = q$ be the unique fixed point of $next$ whose existence is guaranteed by Th. 2. In analogy with a Markov chain, $M^*$ serves as a stationary distribution. As part of our analysis in this section (Cor. 1), we will show that it also serves as a limit distribution. That is, starting from any mass distribution $M_0$ such that $\mathbf{M_0} = \mathbf{M}^*$, $\lim_{t \to \infty} next^t(M_0) = M^*$. However, obtaining $M^*$ by repeated application of $next()$ may take a long time, which has been observed to be an important problem for applying relocation policies in practice [Braverman et al., 2019].

The problem of *dynamic relocation* is to find a sequence of policies that accelerate convergence. Given an initial mass distribution $M_0$, a dynamic relocation policy computes a sequence of policies $\{\pi_i\}_{t=0}$ such that the sequence $\{M_t\}_{t=0}$, where each $M_{t+1} = next_{\pi_t}(M_t)$, converges to $M^*$ faster than the sequence obtained when $next_\pi$ is applied iteratively. Unless no region is saturated, mass distributions $M$ and $M' = next_{\pi'}(M)$ uniquely defines $\pi' = (M' - \mathbb{1}F(M))/\mathbf{E(M)}$ (see Sec. 5). Hence the problem of dynamic relocation can be stated as identifying a sequence $\{M_t\}_{t=0}$ as above that satisfies:

1. mass conservation: for every step $t$, $\mathbf{M_t} = \mathbf{M}^*$

2. relocation constraint: for every step $t \geq 0$, $M_{t+1} \geq \mathbb{1}(F(M_t))$

## 6.2 OUR DYNAMIC RELOCATION POLICY

We now introduce our dynamic policy based on our fixpoint construction. Before doing so, we introduce some additional notation.

- Let $\nu(q)$ be the unique fixpoint of $next_\pi$ with mass $q$. That is, $next_\pi(\nu(q)) = \nu(q)$ and $\boldsymbol{\nu}(\mathbf{q}) = q$.

- Let $\nu^{-1}$ be the $\nu$'s inverse extended to all mass distributions, $\nu^{-1}(M) = \max\{q' : \nu(q') \leq M\}$, that is, captures the mass of the greatest fixpoint vector that is no larger than $M$.

We next define our dynamic relocation policy inductively. For a vector $A$, define $(\![A]\!)_+ = \max(A, \mathbb{0})$, where as usual $\mathbb{0}$ is the all zeroes vector and the maximum is taken coordinate-wise. Given $M_t$, we define $q_{t+1}$ to be the maximum solution to

$$\sum_i (\![\nu(q_{t+1}) - \mathbb{1}F(M_t)]\!)_+[i] = \mathbf{E}(\mathbf{M_t})$$

Th. 1 implies that $\nu(q)$ is monotone increasing and continuous in $q$. Thus, $(\![(\nu(q) - \mathbb{1}F(M_t))]\!)_+$ is also monotone non-decreasing and continuous in $q$. If $\mathbf{E}(\mathbf{M_t}) > 0$, $q_{t+1}$ is the unique solution, and if $\mathbf{E}(\mathbf{M_t}) = 0$, the set of solutions $\{q' \mid \nu(q') \leq \mathbb{1}F(M_t)\}$ has as its maximum $q_{t+1} = \nu^{-1}(\mathbb{1}F(M_t))$, hence $q_{t+1}$ is well defined.

We define $M_{t+1} = \max(\mathbb{1}F(M_t), \nu(q_{t+1}))$. To verify that this is a dynamic policy, note that the relocation constraint, $M_{t+1} \geq \mathbb{1}(F(M_t))$, is satisfied by construction. As for the mass conservation constraint, $\mathbf{M_{t+1}} = \sum_i (\![\mathbb{1}F(M_t) + (\nu(q_{t+1}) - \mathbb{1}F(M_t))]\!)_+[i] = \mathbf{F}(\mathbf{M_t}) + \mathbf{E}(\mathbf{M_t}) = \mathbf{M}^*$.

## 6.3 CONVERGENCE PROPERTIES

We now analyze the convergence properties of our dynamic relocation policy. We show that it converges, at least linearly, to $M^*$ and that a same argument shows that $M^*$ acts as a limit distribution. To obtain a stronger guarantee, we relate the progress made by the dynamic policy to $\mathbf{E}(\mathbf{M})$, the mass of drivers we actually control. The resulting guarantee also provides a bound on the mass of extra empty rides relative to those inherent in $M^*$.

Our first result is that the $q_t$ are strictly increasing at a rate that implies (at least linear) convergence. Let $\Delta(t) = \mathbf{M}^* - q_t$. Since $\nu(\mathbf{M}^*) = M^*$, $\Delta(t)$ can serves as a measure of the distance between the fixed point we have succeeded in "constructing" so far and our goal.

**Lemma 6** $\exists c > 0$ s.t. $q_{t+1} - q_t \geq c\Delta(t)$.

Lem. 6 shows that we are guaranteed to make progress even if we do not control the relocation of any drivers, and is analogous to results about the convergence of Markov chains toward their stationary distributions. Because it does not rely on relocation, it is quite weak; a simpler version of its proof shows that simply statically using $\pi$ satisfies essentially the same bound. This also shows that $M^*$ acts as a limit distribution of the dynamics.

**Corollary 1** *For all mass distributions $M_0$ such that $\mathbf{M_0} = \mathbf{M}^*$, $\lim_{t\to\infty} next_\pi^t(M_0) = M^*$*

To get an (often) stronger bound, recall that the essence of our dynamic relocation policy is that it makes the maximum progress it can given the mass it controls. Let $\Delta^E(t) = \mathbf{E}(\mathbf{M_t}) - \mathbf{E}(\nu(\mathbf{q_t}))$. This captures the mass of relocating drivers beyond that present in $\nu(q_t)$. The following lemma, whose proof is in App. C, shows that this provides a lower bound on the progress made by the algorithm.

**Lemma 7** $q_{t+1} - q_t \geq \Delta^E(t)$.

When $\Delta^E(t)$ is large, which requires many drivers to be relocating, Lem. 7 guarantees that we make rapid progress toward our goal. It also guarantees that our dynamic relocation policy has an attractive welfare property: because we make progress at least equal to the mass of drivers relocating who would not also be relocating under $\nu(q_t)$ (and also under $M^*$), we end up paying the cost of extra relocation at most once for each driver. Thus, the total excess mass of empty rides $\sum_t \mathbf{E}(\mathbf{M_t}) - \mathbf{E}(\mathbf{M}^*)$ is at most $M^*$, as the following observation, whose proof is in App. C, shows.

**Observation 3** $\sum_t (\mathbf{E}(\mathbf{M_t}) - \mathbf{E}(\mathbf{M}^*)) \leq \mathbf{M}^*$.

We are aware of no prior approach that provides such a guarantee. It is easy to see, for example, that the static policy $\pi$ can have more than one extra relocation per driver. (Construct an example where all the drivers start in the same region and essentially all relocate in the first step but this is not the fixpoint.) Many other prior approaches are heuristic and do not even provide a guarantee of convergence.

## 7 SIMULATIONS

Our simulations are based on a dataset from Didi from 2016 for an unspecified region in China that was processed by Braverman et al. [2019] into a form suitable for our model, representing nine major regions of the city. For completeness we provide the model parameters in App. A. Notably, these include non-uniform distances between regions. While our theoretical analysis assumed unit travel times, our constructive approach is easily adapted to this richer setting. See App. A for a discussion. Braverman et al. [2019] perform their experiments in a large continuous-time system while we apply our discrete time model using 15-minute intervals. Our results are similar which validates our discrete approach. The results of both this recreation and a variant we introduce show that our dynamic relocation policy outperforms prior approaches in this setting.

## 7.1 WHAT WE COMPARE

We compare our approach, which we refer to as CON, with four other policies:

STA. This is a static policy that sets $\pi_t = \pi$ for all $t$. From Cor. 1 it follows that STA guarantees convergence to the fixpoint, yet, as we pointed out, it may do so slowly. Thus STA represents a baseline in the absence of a more sophisticated dynamic policy.

GDY. This is a greedy policy that distributes the relocating mass proportional to the unmet demand in each region with a one-step look ahead. That is, it takes $\pi_t[i] \propto \left( \llbracket W\mathbb{1} - \mathbb{1}F(M_t) \rrbracket \right)_+$, which guarantees that as many relocating drivers as possible will have a passenger at time $t + 1$ while spreading them among the regions where they can be useful. As GDY does not depend on $\pi$, it may not converge to the fixpoint, but it does provide a meaningful baseline for other metrics based purely on the provision of service.

LKA. This is a heuristic proposed by Braverman et al. [2019] that forms a model based on the average of demand patterns over a lookahead window and finds the optimal static policy for this average. For the look head, we use 2, 3, and 4 steps.

HMR. We adapt the dynamic policy of Hosseini et al. [2021], which dispatches a single car at a time, to our setting. In particular, their algorithm computes a measure of which region will generate the most long-run service and sends the car there. Since the results of this computation do not change until a region is saturated, we adapt their policy by assigning relocating drivers to this region until (a) it becomes saturated or (b) the mass reaches the fixpoint mass of the region.

## 7.2 RESULTS

Our first experiment is based on a similar experiment by Braverman et al. [2019] and uses the same parameters. We simulate the system for four hours. Initially the supply of drivers is large relative to the demand for rides, so almost all requests can be served. However, after two hours demand increases sharply and passengers' destinations (i.e. $V$) are randomly permuted. Table 1 show the resulting availability averaged over 20 runs. Availability is a standard metric in this setting and is the average fraction of demand served across regions. (See App. B for more discussion of this and other metrics.)

Our results for STA and LKA are qualitatively similar to those reported by Braverman et al. [2019] in their Table 4. In the first two hours almost all demand is served while the STA performance drops with the demand transition as it is slow to adapt to the new demand pattern. In contrast, LKA with lookaheads of 2, 3, and 4, steps (30, 45, or 60

|  | Hour1 | Hour2 | Hour3 | Hour4 | 4-hour Total |
|---|---|---|---|---|---|
| CON | 1.000 | 1.000 | 0.937 | 0.938 | 0.969 |
| GDY | 1.000 | 1.000 | 0.912 | 0.915 | 0.957 |
| HMR | 1.000 | 1.000 | 0.888 | 0.918 | 0.952 |
| LKA4 | 0.983 | 0.990 | 0.897 | 0.931 | 0.950 |
| LKA3 | 0.982 | 0.991 | 0.882 | 0.930 | 0.946 |
| LKA2 | 0.983 | 0.995 | 0.858 | 0.929 | 0.941 |
| STA | 0.983 | 0.998 | 0.814 | 0.929 | 0.931 |

Table 1: Availability improvement in a 4 hour simulation with single demand change

minutes) can anticipate the change and has a smoother transition. The other three approaches (HMR, GDY, and CON) do not anticipate the change but simply adapt to it rapidly. HMR and GDY achieve similar performance to LKA, while our CON adapts the fastest and outperforms the alternatives.

Our second experiment extends this approach by keeping the overall level of demand ($W$) fixed while permuting the pattern of demand as in the previous experiment. We permute the demand 19 evenly-spaced times over the course of the experiment and report results averaged over 20 runs. We treat STA as a baseline and report the percentage change in availability relative to it for each of the other policies. Fig. 3 shows two variations. On the left, we fix the ratio between supply ($M$) and demand ($W$) at 1.15 and vary the number of steps between demand changes. We start this variation from 5 steps, which corresponds to updates every 75 minutes, as we think smaller values represent unreasonably fast changes of demand. In contrast to our first experiment, LKA only slightly outperforms STA. Intuitively, while LKA can anticipate future demand patterns without a significant excess of drivers, its ability to reposition them for future demand while still serving current demand is limited. Since CON simply adapts changing demand patterns rapidly, it remains highly effective although the benefits decrease as changes in demand become less frequent. Neither GDY nor HMR performs well in this setting relative to STA.

On the right, we fix the number of steps between demand changes at 10 and vary the ratio between supply and demand. The performance of all policies improves relative to STA as the supply of drivers increases. GDY, as a simple baseline, is still outperformed by the other approaches. However, now HMR does better than the lookahead approaches at sufficiently high levels of supply.

## 7.3 ADDITIONAL SIMULATIONS

In App. B we present the results additional simulations on synthetic data. They show that with static demand patterns CON consistently converges substantially faster than other policies and has a performance that is often close to or

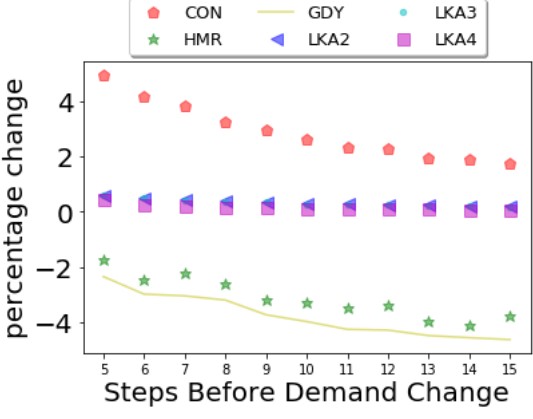 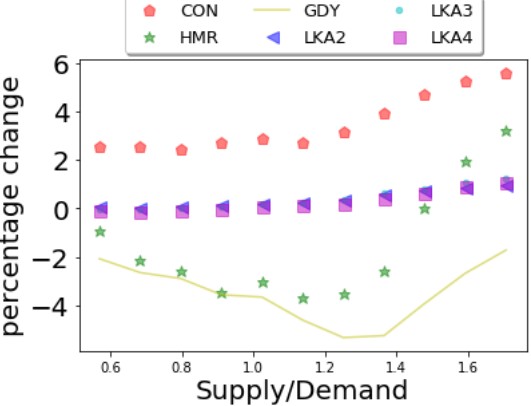

Figure 3: Comparison of Policies with STA

matching a lower bound. The effect of this on efficiency (the mass of passengers served) relative to the other policies is, however, quite small. When targeting an objective for selecting an optimal fixedpoint that puts weight on fairness rather than just efficiency, CON leads to economically meaningful improvements in availability, showings its ability to target a wider range of objectives than previous approaches.

## 8 DISCUSSION

We have studied a model of relocation policies for ridesharing platforms and given a constructive characterization of the unique fixpoint of system dynamics. Using this construction, we designed a dynamic policy that provides guarantees about its rate of convergence to the fixpoint and analyzed the magnitude of these benefits in simulations.

To obtain these results, we used a stylized model. We conclude by discussing the extent to which our results extend to richer models. Within our basic setup we made use of several assumptions. First, that drivers do not relocate if there are waiting passengers. This is typically a mild assumption in models of spatial demand imbalances, but would be more relevant in a study of temporal ones [Ma et al., 2019]. Second, that drivers carry passengers regardless of their destination. This is a common requirement, but some work has investigated the benefits from allowing strategic passenger selection [Afeche et al., 2018]. Third, that there is a positive demand in in between every two regions. This assumption guarantees ergodicity, and in many of our results could likely be reduced to that weaker requirement. It is also relevant for our convergence rate analysis of dynamic policies, although there it could likely be replaced by an eigenvalue-based bound, as is typical in the analysis of mixing time of Markov chains. This assumption seems quite reasonable in practice: All it requires is that, for a reasonable decomposition of a city to regions, people occasionally wish to travel from any given part of the city to

any other. Unless this grid of regions is very fine this seems quite likely. Fourth, that the relocation policy is the same for every region. We believe our results can be obtained without it, albeit with substantial additional notational clutter. For our experiments on the Didi data, our results demonstrate empirically the effectiveness of this extension.

There are also a number of features our basic model excludes. We work with a continuum of drivers, but previous work has shown that dynamics with a continuum of drivers approximately hold with discrete drivers as the number of drivers grows [Braverman et al., 2019, Banerjee et al., 2017]. We also assume that time is discrete and all journeys take a single unit of time. As in the simulations, discrete but non-uniform journey times can be modeled by introducing additional "regions" which represent drivers in transit, and our results likely generalize.[1] Adding travel times to the linear programming approach is straightforward, and previous work has observed that, as long as the platform controls the relocation policy, non-uniform travel times do not significantly affect the behavior of the model [Bimpikis et al., 2019]. Journey times of a single pair of regions may vary over time. A key takeaway from prior work is that as long as the number of vehicles is large this factor is essentially irrelevant, which is why we did not include it in our model or experiments [Braverman et al., 2019].

While we do not explicitly consider a dynamic willing demand matrix $W$, it is the motivation for our results in Sec. 6.3 with guarantees on convergence speed towards the fixed point comes from this issue. In particular, policies that are slow to adapt can be stuck, never approaching the fixpoint before $W$ changes. In contrast, our dynamic policy can deal with a changing demand matrix by converging rapidly before the change is so large that the target becomes outdated. Our simulations on synthetic data (the left subfigures of Fig. 1, Fig. 2, Fig. 4, and Fig. 6 in the appendix) show that

---

[1]This would involve relaxing the previously discussed positive demand assumption.

the convergence towards the fixed point is indeed very fast and is generally faster than an expected substantial change of the demand. Table 1, which is drawn from real scenarios, shows similarly rapid adaptation (note the availability in hour 3, after the change in W, is nearly identical to hour 4 for CON while most other methods do substantially better in hour 4 than hour 3.) Both subfigures of Fig. 3 also deal with dynamic W and the results demonstrate the superior performance of our dynamic policy.

Our constructive algorithm myopically attempts to make as much progress toward the fixpoint as possible. An interesting direction for future work would be to treat the problem of dynamic relocation as a planning or reinforcement learning problem. Our theoretical results can be viewed as providing a characterization of key aspects of a model to enable model-based approaches. One could imagine adding an approach based on Monte Carlo Tree Search or similar techniques [Sutton and Barto, 2018]. From this perspective, the lower bound (LB) we compare to in some of the additional experiments in the appendix is the optimal plan for a relaxed version of the problem and in those experiments CON shows near-optimal performance.

Finally, we do not model prices or monetary relocation costs. As these do not affect the system dynamics, most of our results would be unchanged. The primary effect would be to adjust the objective of problem OA (Fig. 2) accordingly. Nor do we consider settings where drivers have partial or total control over relocation decisions, and thus the sequential decision problem is faced by drivers rather than the platform. Of course, a major role of prices is to influence the relocation decisions of rational drivers. Our results provide key characterizations that justify formulating the problem from the perspective of a single driver in terms of the fixpoint induced by the behavior of other drivers, an approach which has been fruitful in other game-theoretic work on marketplaces [Kash et al., 2015].

## Acknowledgements

This material is based upon work supported by the National Science Foundation Program on Fairness in AI in collaboration with Amazon under Grant No. 1939743, by the National Science Foundation under Grant No. 1918429, and by the Discovery Partners Institute (DPI) Science Team Seed Grant Program. DPI is part of the University of Illinois System.

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
