# OpenReview forum: "Dynamic Relocation in Ridesharing via Fixpoint Construction"
_auai.org/UAI/2022/Conference — UAI 2022 Poster_

### Official Review · Reviewer_W2sS · 2022-04-09

**Q2(1) Originality/Novelty:** 3
**Q2(2) Significance/Impact:** 1
**Q2(3) Correctness/Technical Quality:** 3
**Q2(6) Clarity Of Writing:** 4
**Q6 Overall Score:** 4
**Q8 Confidence In Your Score:** 4

**Q1 Summary And Contributions:**

This paper proposes a strategy for dynamic driver relocation in a simplified problem setting. The paper offers proofs of convergence guarantees which several previous works lack and shows empirical performance improvement in convergence time, availability, and efficiency in several simulated settings.

**Q2 Assessment Of The Paper:**

More detailed information regarding each of these aspects is given below:

**Q2(4) Quality Of Experiments (Optional):**

3: Good: The experimental evaluation is adequate, and the results convincingly support the main claims.

**Q2(5) Reproducibility:**

2: Fair: Key resources (e.g., proofs, code, data) are unavailable but key details (e.g., proof sketches, experimental setup) are sufficiently well-described for an expert to confidently reproduce the main results.

**Q3 Main Strengths:**

This paper proposes a clean and theoretically grounded approach for dynamic driver relocation. The empirical experiments designed are suitable for the claims made by the paper. The paper is clearly written, uses consistent notation, and easily understood.

**Q4 Main Weakness:**

1. My main concern about this paper is its relevance to AI. The paper does not attempt to relate to the UAI community at all and therefore may have limited impact in this regard.
2. The paper does not attempt to compare to any (possibly learning-based) baselines that would commonly be used by the UAI community. This could be due to the lack of overlap between this problem domain and the UAI community, but it should be feasible to design such a baseline for this problem setting. Doing so could help alleviate the above concern as well.
3. Another concern is the simplified problem setting considered, which are quite simplified. While the authors discuss richer models in the Discussion section, there could be more experiments addressing these regards. I'm particularly interested in the non-uniform journey time setting.

**Q5 Detailed Comments To The Authors:**

A few clarifications regarding the stated weaknesses above
1. Dynamic driver relocation is a sequential decision problem, and it's conceivable that planning or reinforcement learning-based methods could be used here. These methods could be more familiar to the UAI community.
2. The cited work Iglesias 2019 considers a much more extensive problem setting. While the authors of this paper use a simplified setting for developing a theoretically grounded algorithm, it would be helpful to see empirical comparisons in a more extensive problem setting.

**Q7 Justification For Your Score:**

While the proposed driver relocation strategy is clean and theoretically grounded, this paper does not do enough to demonstrate relevance to the UAI community. The problem domains considered is also quite simple, while some cited baseline works offer more compelling experimental domains.

**Q9 Complying With Reviewing Instructions:**

1: Yes.

---

### Official Review · Reviewer_RHYa · 2022-04-10

**Q2(1) Originality/Novelty:** 3
**Q2(2) Significance/Impact:** 2
**Q2(3) Correctness/Technical Quality:** 3
**Q2(6) Clarity Of Writing:** 4
**Q6 Overall Score:** 7
**Q8 Confidence In Your Score:** 3

**Q1 Summary And Contributions:**

The authors study a model for dynamic relocation for drivers in a ride-sharing environment. By means of a simplified model, authors show that there is a unique fix-point of the system dynamics. In line with their study, they design a policy with guarantees on the convergence properties. Moreover, they show experimentally that these guarantees are better than prior work.

**Q2 Assessment Of The Paper:**

More detailed information regarding each of these aspects is given below:

**Q2(4) Quality Of Experiments (Optional):**

4: Excellent: The experimental evaluation is comprehensive and the results are compelling.

**Q2(5) Reproducibility:**

2: Fair: Key resources (e.g., proofs, code, data) are unavailable but key details (e.g., proof sketches, experimental setup) are sufficiently well-described for an expert to confidently reproduce the main results.

**Q3 Main Strengths:**

Novelty: In contrast to previous work, authors provide a rigorous mathematical analysis of the problem and study its properties, focusing on the system dynamics and the convergence to a unique fix-point.

Significance: Their study is performed for an arbitrary policy, which means that the conclusions drawn from it are general for future relocation approaches. The results show improvement, specially in terms of convergence, with respect prior work.

Quality of experiments: The experimental analysis is complete and adopts some of the experimental setup performed in previous work, hence resulting in a fair comparison. The main claims can be easily understood by looking at the results presented.

**Q4 Main Weakness:**

Significance: Despite the proofs and results are general, authors should discuss how future work can include their results and benefit from them.

**Q5 Detailed Comments To The Authors:**

How your approach would adapt if you have incoming demands at each time-step? Are the results you show still valid in that situation?

**Q7 Justification For Your Score:**

The most valuable traits of this work are novelty, due to the rigurous mathematical anaysis and the study of the system dynamics and fixpoint, and significance, due to its generality and improvement with respect to prior work.

**Q9 Complying With Reviewing Instructions:**

1: Yes.

---

### Official Review · Reviewer_cfDb · 2022-04-12

**Q2(1) Originality/Novelty:** 2
**Q2(2) Significance/Impact:** 2
**Q2(3) Correctness/Technical Quality:** 3
**Q2(6) Clarity Of Writing:** 2
**Q6 Overall Score:** 5
**Q8 Confidence In Your Score:** 1

**Q1 Summary And Contributions:**

This paper studies how a ridesharing platform can relocate drivers between different regions of a service area so as to maximize the average fraction of full rides. A stylized model, resembling in some aspects a Markov chain, is proposed. The determination of an optimal relocation policy is formulated as a linear program. A dynamic relocation policy via fixed point construction is also studied. The paper concludes with simulations on a dataset corresponding to an unspecified area in China.

**Q10 Ethical Concerns (Optional):**

No ethical concern.

**Q2 Assessment Of The Paper:**

More detailed information regarding each of these aspects is given below:

**Q2(4) Quality Of Experiments (Optional):**

3: Good: The experimental evaluation is adequate, and the results convincingly support the main claims.

**Q2(5) Reproducibility:**

2: Fair: Key resources (e.g., proofs, code, data) are unavailable but key details (e.g., proof sketches, experimental setup) are sufficiently well-described for an expert to confidently reproduce the main results.

**Q3 Main Strengths:**

* The technical content seems non-trivial.
* The position of the contribution w.r.t. the related work is precisely argued.
* The theoretical contributions are accompanied by numerical experiments on real-world data.

**Q4 Main Weakness:**

* The paper is hard to read; no illustrative examples are presented, and the abundance of notations sometimes hinders the reading.

**Q5 Detailed Comments To The Authors:**

It is first noted that, if the same relocation policy is followed at each timestep, then the system converges toward a fixpoint distribution of drivers across the regions. An algorithm is proposed that, given a relocation policy, computes the fix-point distribution of drivers across the regions. It is shown that this fixpoint is unique.

For a ridesharing platform that has some control over the relocation policy, a linear program is formulated for determining a policy that maximizes the number of full rides.

The dynamic relocation problem consists then in finding a sequence of policies that allows to speed up the convergence. An approach is proposed for this purpose.

Finally, the proposed method is compared with baseline methods and state-of-the-art methods on real-world data. In particular, the ability to quickly converge to a new fixpoint when the matrix of demands changes is compared with other methods.

Typos
- The references souhld be put between parentheses or brackets (in all the paper).
- p.3: the vector of the available drivers available each location -> the vector of the available drivers at each location.
- p.3: proportional to $\sigma(T_M)$ -> proportionally to $\sigma(T_M)$?
- p.5: the fixed point of of -> the fixed point of.
- p.7: that that forms a model -> that forms a model.
- p.8: targeting a objective -> targeting an objective.

**Q7 Justification For Your Score:**

The proposed model has the advantage of being based on theoretical foundations that seem solid, but I am not an expert on the topic and the writing of of the paper, while rigorous, did not allow me to form a definite opinion about that contribution.

**Q9 Complying With Reviewing Instructions:**

1: Yes.

---

### Official Review · Reviewer_nf6d · 2022-04-15

**Q2(1) Originality/Novelty:** 2
**Q2(2) Significance/Impact:** 2
**Q2(3) Correctness/Technical Quality:** 3
**Q2(6) Clarity Of Writing:** 3
**Q6 Overall Score:** 5
**Q8 Confidence In Your Score:** 3

**Q1 Summary And Contributions:**

The paper presents a coarse-grained approach for studying and mitigating the imbalance of drivers in urban areas when taxis and ridesharing cars operate.


**Q2 Assessment Of The Paper:**

More detailed information regarding each of these aspects is given below:

**Q2(4) Quality Of Experiments (Optional):**

3: Good: The experimental evaluation is adequate, and the results convincingly support the main claims.

**Q2(5) Reproducibility:**

2: Fair: Key resources (e.g., proofs, code, data) are unavailable but key details (e.g., proof sketches, experimental setup) are sufficiently well-described for an expert to confidently reproduce the main results.

**Q3 Main Strengths:**

The proposed model is rather abstract and simple, in order to capture the most essential elements of car relocation.
The proposed dynamic relocation algorithm performs well compared with some alternatives.

**Q4 Main Weakness:**

The significance of the proposed model is unclear.
Some aspects of the model could be better discussed.

**Q5 Detailed Comments To The Authors:**

One general aspect that is not fully clear is whether the three "key properties" listed in Section 1 could be established also using other models. Without this discussion, it is hard to assess the value of the proposed model, beyond the fact that it is simple and does not consider several details that are included in other models.
More specifically, why cannot the study of fixed points be carried out also using other more complex models? What are the features of such models that prevent this study? Do more complex models allow deeper analysis?

The proposed model is described in Section 2.2.
- The demand matrix W is considered given but, in real applications, it dynamically changes over time (e.g., consider commuters and rush hours). How is this aspect accounted for in the authors' model? Ignoring the dependence of W from t means that the steady-state reached by the system corresponds to a possibly outdated demand W.
- Another strong limitation is the strict positivity of demand between any two regions, and the authors correctly discuss this issue in Section 8.

Comments on minor issues and on typos follow.
Please use parenthesis for references (\citep{}?).
Page 3: the available drivers available each location (?)
Page 5: of of
Page 6: solution,and -> solution, and
Page 6: the last sentence of the right column discusses "Con", which is introduced only later on
Page 7: Hmr,Gdy -> Hmr, Gdy
Page 7: performance well -> perform well
Page 8: a objective -> an objective
Page 8: in in
Some references miss volume and pages (some papers on journals), miss venue ([Waserhole and Jost, 2012])

**Q7 Justification For Your Score:**

The idea of using an abstract model to elucidate the most basic properties of a domain is potentially interesting. The approach seems sound but the paper misses a discussion of the actual value of the proposed approach with respect to other approaches.
The paper is overall clear and experiments are adequate to show that the proposed algorithm for dynamic relocation works well.

**Q9 Complying With Reviewing Instructions:**

1: Yes.

---

### Decision · Program_Chairs · 2022-05-15

**Decision:**

Accept (Poster)

**Comment:**

Meta Review: The authors have cleared most of the reviewers' doubts during the rebuttal, after which all reviewers are agreeing to accept the paper (reviewer W2sS did not officially updated their score to 6 as indicated in the reply).